# Architectural Conditioning for Disentanglement of Object Identity and Posture Information

**Kazutoshi Sagi, Takahiro Toizumi & Yuzo Senda**
Data Science Research Laboratories
NEC Corporation
Kawasaki, Kanagawa 211-8666, JAPAN
`ksagi@ah.jp.nec.com`
`t-toizumi@ct.jp.nec.com`
`y-senda@bc.jp.nec.com`

## Abstract

This paper proposes an architectural conditioning approach for disentanglement of object identity and posture information. Challenging in deep learning is to disentangle a specific condition from learned representations. The proposed architectural conditioning employs a rigid matrix operation as a layer in an autoencoder to achieve disentangling of a specific condition. This paper demonstrates how the proposed conditioning learns rotation-invariant representations. Using the architectural conditioning, rolling of latent vectors corresponds to rotation of an object in an image. Thus the object posture information is positionally represented in a latent vector. The experimental results on MNIST and 3D chair model images show that this conditioning enables networks to learn rotational bases as their weights. An arbitrary view can be inferred using different views.

## 1 Introduction

Recent deep neural networks (DNNs) have acquired invariant representations in various computer vision tasks. A capsule network (Sabour et al., 2017; Geoffrey E Hinton, 2018), one of the latest proposed architectures, uses vectors as groups of nodes. Even a shallow capsule network shows the performance comparable to deep convolutional networks. However it is still challenging to identify which specific features contribute to represent a single condition. The disentangling of representations is a key study in practical problems especially in a case of small labelled dataset due to its high sensitivity to limited conditions.

Many studies use deep generative models to learn the disentanglement and to obtain modality-invariant representations. Kingma et al. (2014); Suzuki et al. (2017); Vedantam et al. (2018) modified the objective of their models by considering a joint probabilistic distribution of images and attributes. Cohen & Welling (2016; 2017) implemented a physics-based coordinate transformation into representations to steer them. Yang et al. (2015) disentangled identity and viewpoint units using out-of-plane rotation images by combination of an autoencoder and a recurrent neural network with curriculum learning. However their architectures tend to be complex in implementations. We think that a network architecture is supposed to be simpler by exploiting flexibility of DNNs.

We propose an architectural conditioning approach, which employs one rigid matrix operation as a layer in an autoencoder. The layer controls paths of latent variables between its encoder and decoder to make the network learn a condition as a matrix operation. The architectural conditioning has already been demonstrated in an application for an un-seen view target recognition in synthetic aperture radar images, which images are highly sensitive to an observational viewing change (Sagi et al., 2018). The purpose of this paper is to verify the proposed conditioning performance on 2D image and 3D object rotations in generally used images.

## 2    PROPOSED METHOD

This paper shows use of the architectural conditioning to obtain rollable latent space (RLS). The RLS is a rotation-invariant representation, which is interchangeable with different viewing of an object. In the RLS, a latent vector consists of multiple sub-vectors, each of which has a fixed length of structural and directional information. The number of sub-vectors correspond to the number of equally spaced viewing directions. The sub-vector can be rolled to represent a rotated image of its structural information. An RLS encoder converts an image $\mathbf{X}$ seen from a direction $\theta_i$ to a latent vector $\mathbf{Z}$; $\mathbf{Z}(\theta_i) = Encoder(\mathbf{X}(\theta_i))$. $\mathbf{Z}$ from another direction $\theta_j$ can be approximated by simply applying roll function to $\mathbf{Z}$,

$$\mathbf{Z}(\theta_j) = Encoder(\mathbf{X}(\theta_j)) \tag{1}$$
$$\simeq Roll(\mathbf{Z}(\theta_i), j - i), \tag{2}$$

where $Roll(\mathbf{Z}, s)$ rolls all sub-vectors of $\mathbf{Z}$ by shift parameter $s$.

In a simple RLS, $Roll(\mathbf{Z})$ can be given as $\mathbf{R}^s \cdot \mathbf{Z}_k$, where $\mathbf{Z}_k$ and $\mathbf{R}$ are a sub-vector of $\mathbf{Z}$ and a cyclic permutation matrix, respectively. The roll function interpolatively works to support continuous rotation.

To let $Encoder$ learn this disentanglement, an autoencoder approach is employed. An RLS autoencoder is defined as

$$\mathbf{X}(\theta_j) \simeq Decoder(Roll(Encoder(\mathbf{X}(\theta_i)), j - i)). \tag{3}$$

By feeding images of known objects to the autoencoder with varying $s$, $Encoder$ and $Decoder$ are obtained.

## 3    EXPERIMENTAL RESULTS AND DISCUSSIONS

### 3.1    DISENTANGLING 2D IMAGE ROTATION

An experiment of disentangling 2D image rotation is performed on MNIST images (LeCun, 1998). The MNIST dataset has 60K and 10K images for training and testing respectively. Figure 1 (a) shows the structure of an RLS-based autoencoder used in this study. To promote disentanglement, variational autoencoder is employed (Kingma & Welling, 2013). The encoder and the decoder just consist of one hidden fully connected layer with ReLU activation for each. The number of the latent space dimentions is given as 24, which corresponds to 2 dimensions in 12 viewing directions.

**Rotational Basis Acquisition:** Reconstructions of the test dataset are illustrated in figure 1 (c). The left column and its next column of each row are an input image and a reconstructed image, respectively. The other images are generated in any rotation angles by equation 3 while keeping important details. In a case of 2D image rotation, our model learns rotational basis such as Zernike moments (Khotanzad & Hong, 1990). Figure 1 (b) shows visualization of the encoder weight, which maps image space of 784 dimensions to latent space of 24 dimensions. The weight corresponding to each latent variable is reshaped into $28 \times 28$ basis in the figure. It is obvious that those bases are point-symmetric patterns. Variation of basis, i.e. similar features with zernike polynomials, increases by determining more dimensions in each viewing direction.

### 3.2    DISENTANGLING 3D OBJECT ROTATION

Compared to the 2D image rotation, learning disentanglement of 3D object rotation is more difficult due to self-occlusion of an object in an image; information from the object backside is never retrieved using one single image. This study evaluate the proposed conditioning for disentangling 3D object rotation using images rendered from chair CAD models provided by Yang et al. (2015). The original chair model dataset was provided by Aubry et al. (2014) and contains 1393 different chairs. However a subset of 809 chair models are selected in order to remove near duplicate models and low-quality models by following Dosovitskiy et al. (2015). The first 500 models are used as a training set and the remaining 309 models are used as a test set. Each chair model is rendered from 31 azimuth angles and 2 elevation angles ($20°$ and $30°$) at a fixed distance to the virtual camera and cropped into $64 \times 64 \times 3$ pixels. In our experiment images of the $20°$ elevation angle are used. Figure 2 (a) is the structure of an RLS-based autoencoder for the experiment. A deep convolutional

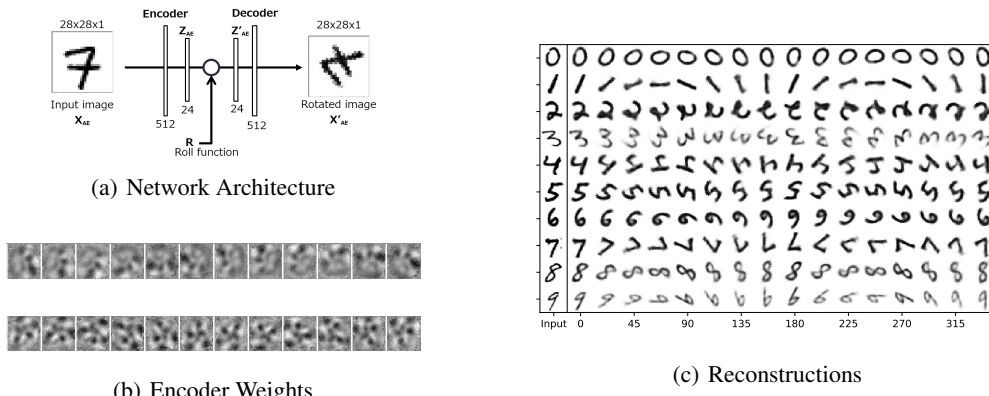

(a) Network Architecture

(b) Encoder Weights

(c) Reconstructions

Figure 1: (a): A network architecture used in the experiment of 2D image rotation. (b): Visualization of the trained encoder weight. The weight corresponding to each latent variable is reshaped into $28 \times 28$ basis. (c):Reconstructions of the test dataset. An input and reconstructions in given rotation angles generated by equation 3 are presented from the left column of each row.

encoder-decoder architecture proposed by Yang et al. (2015) is applied to our autoencoder architecture. The encoder network consists of three $5 \times 5$ convolution layers with stride 2 and 2-pixel padding and ReLU activation, and two fully connected layers. The decoder takes a symmetric architecture to the encoder. Fixed stride-2 convolution and upsampling are employed instead of max-pooling and unpooling. The number of the latent space dimensions is given as 992, which corresponds to 32 dimensions in 31 viewing directions.

The proposed conditioning helps the network to link information of different views each other. Results of reconstructions are shown in figure 2 (b). The generated chair images in any rotation angles capture common features such as arm rests, legs and a seat back. Hence the object posture information is positionally conserved in a latent vector as well as the 2D rotation experiment.

**Class Interpolation:** Our model can generate novel chair classes by interpolating between two given chairs in the latent space. Using their latent vectors $\mathbf{Z}_1$ and $\mathbf{Z}_2$, the interpolated latent vector $\mathbf{Z}_{interp}$ is calculated by $\mathbf{Z}_{interp} = \beta \mathbf{Z}_1 + (1 - \beta) \mathbf{Z}_2$. Examples of class interpolation between two chairs are presented in figure 2 (c). The interpolated chairs indicate smooth transformations between the pair of input chairs. The interpolated vectors are also ruled by equation 2. Those facts imply that the learned latent space is able to represent unspecified or missing chair images.

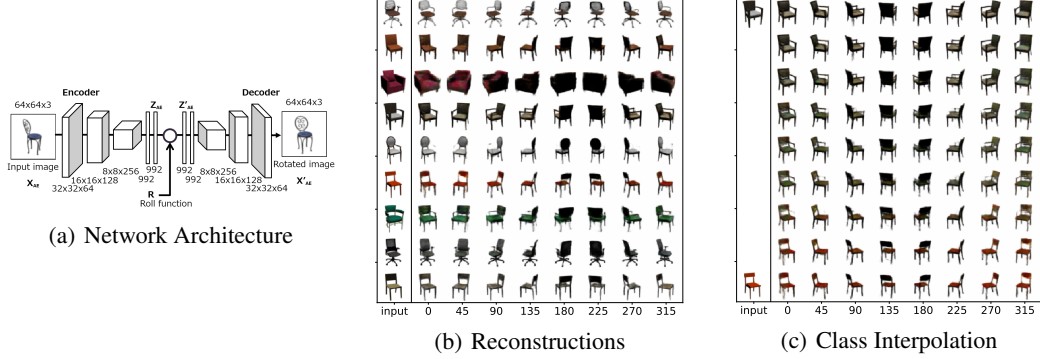

(a) Network Architecture

(b) Reconstructions

(c) Class Interpolation

Figure 2: (a): A network architecture used in the experiment of 3D object rotation. (b): Reconstructions of the test dataset. An input and reconstructions in given rotation angles are shown from the left column of each row. (c): Class inter-transformations. Two inputs are seen at left top and left bottom corners. Following chairs are generated using the interpolated latent vectors by equation 2.

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
