# OpenReview forum: "Architectural Conditioning for Disentanglement of Object Identity and Posture Information"
_ICLR.cc/2018/Workshop — Reject_

### Official Review · AnonReviewer3 · 2018-03-09
**Accept**

**Rating:** 7
**Confidence:** 3

**Review:**

This paper introduces an architectural conditioning for object identity and pose disentangling. The method is sound and the results are expressive, even though mostly on the toy datasets.

The contribution is on par with the workshop track and should be accepted.

---

### Official Review · AnonReviewer1 · 2018-03-09

**Rating:** 4
**Confidence:** 4

**Review:**

The paper demonstrates that a “rolling representation” can be used in an encoder-decoder network architecture to generate views of object. More specifically, the authors encode an image of an object into a vector, “roll” (cyclically permute) this vector and apply the decoder to it, and train the system to generate a different view of an object, with view difference proportional to the amount of “roll”.

Pros:
- It is an interesting fact that such a rolling architecture works.

Cons:
- Novelty is limited: the idea has already been presented in (Sagi et al., IGARSS 2018 ) (for radar images) - apparently, by the same authors
- Compared to the aforementioned paper, there is not much contribution. The authors just show that the proposed encoder-decoder works on two other datasets.
- The approach seems to be fully supervised, requiring knowing the poses corresponding to the images.
- There is no motivation of why exactly is this approach specifically interesting. Does it perform better than alternatives? Does it allow for something that previously was impossible?

To conclude, the paper looks somewhat interesting, but it’s a minimal addition to a previously published paper. It's unclear why this addition is interesting. Therefore I believe the paper cannot be accepted.

---

### Official Review · AnonReviewer2 · 2018-03-12
**Spacial case of previous works**

**Rating:** 3
**Confidence:** 4

**Review:**

To me this paper seems like a spacial case (for rigid rotation) of existing works. Specifically https://papers.nips.cc/paper/5851-deep-convolutional-inverse-graphics-network.pdf and https://papers.nips.cc/paper/5951-learning-to-linearize-under-uncertainty. Both of these papers propose algorithms which can achieve disentangled representations on datasets generated by arbitrary transformations. Interestingly neither paper is cited.

---

### Decision · Program_Chairs · 2018-03-20
**ICLR 2018 Workshop Acceptance Decision**

**Decision:**

Reject

**Comment:**

Based on the reviews, this paper has not been accepted for presentation at the ICLR workshop. However, the conversation and updates can continue to appear here on OpenReview.